# SFESS: SCORE FUNCTION ESTIMATORS FOR $k$-SUBSET SAMPLING

**Klas Wijk**[1,2]   **Ricardo Vinuesa**[1,2]   **Hossein Azizpour**[1,2,3]

[1]KTH Royal Institute of Technology
[2]Swedish e-Science Research Centre
[3]Science for Life Laboratory
`kwijk@kth.se  rvinuesa@mech.kth.se  azizpour@kth.se`

## ABSTRACT

Are score function estimators a viable approach to learning with $k$-subset sampling? Sampling $k$-subsets is a fundamental operation that is not amenable to differentiable parametrization, impeding gradient-based optimization. Previous work has favored approximate pathwise gradients or relaxed sampling, dismissing score function estimators because of their high variance. Inspired by the success of score function estimators in variational inference and reinforcement learning, we revisit them for $k$-subset sampling. We demonstrate how to efficiently compute the distribution's score function using a discrete Fourier transform and reduce the estimator's variance with control variates. The resulting estimator provides both $k$-hot samples and unbiased gradient estimates while being applicable to non-differentiable downstream models, unlike existing methods. We validate our approach experimentally and find that it produces results comparable to those of recent state-of-the-art pathwise gradient estimators across a range of tasks.

## 1 INTRODUCTION

Subsets are essential in tasks such as feature selection (Balın et al., 2019; Huijben et al., 2019; Yamada et al., 2020), optimal sensor placement (Manohar et al., 2018), learning to explain (Chen et al., 2018), stochastic $k$-nearest neighbors (Grover et al., 2019), and system identification (Brunton et al., 2016). Therefore, understanding and effectively manipulating subsets is an important step in improving machine methods that model discrete phenomena.

A cornerstone of modern machine learning is efficient learning, typically achieved through differentiable models optimized via stochastic gradient descent. However, not all operations useful in modeling are differentiable, necessitating gradient estimation to be compatible with gradient-based optimization. For instance, discrete sampling, including $k$-subset sampling, is not amenable to the reparametrization trick (Kingma & Welling, 2014).

Gradient estimation for Bernoulli and categorical distributions has been extensively studied (Bengio et al., 2013; Jang et al., 2017; Maddison et al., 2017; Dimitriev & Zhou, 2021; De Smet et al., 2023; Liu et al., 2023). These distributions are less structured than subset distributions and do not share their combinatorially large support. A Bernoulli distribution has a support size of 2, a categorical $n$, and a $k$-subset $\binom{n}{k}$. Still, the methods employed in their optimization serve as a blueprint for more structured distributions. Existing approaches for differentiable subset sampling (Xie & Ermon, 2019; Ahmed et al., 2023; Pervez et al., 2023) use either approximate pathwise estimators or relaxed sampling. While these methods are effective, they produce biased estimates and relaxed samples respectively (see Figure 1). This paper seeks to address these limitations by revisiting score function estimators (Glynn, 1990; Williams, 1992; Kleijnen & Rubinstein, 1996), a technique well-established in reinforcement learning (Sutton et al., 1999) and variational inference (Ranganath et al., 2014), but overlooked for subset sampling. In this work, we cover the aforementioned research gap by posing the following question:

> *Can score function estimators compete with approximate and relaxed pathwise gradient estimators in $k$-subset sampling despite their weaker assumptions?*

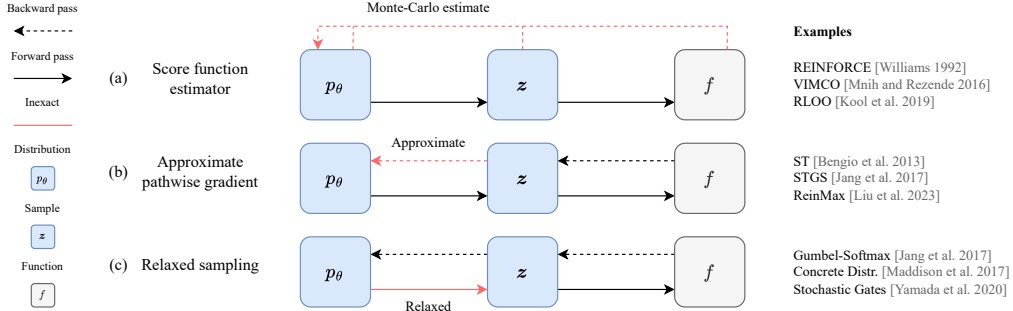

Figure 1: **Gradient estimation for discrete distributions**. Three prominent approaches to gradient estimation for discrete distributions: (a) approximate score function estimator, (b) pathwise gradient estimator, and (c) relaxed sampling. The examples listed estimate the gradients of Bernoulli samples, categorical samples, or both. We propose a score function estimator for $k$-subset sampling to complement existing methods based on approximate pathwise derivatives and relaxed sampling (see Section 5). Because it does not use the pathwise gradient, it is applicable in cases when $f$ is non-differentiable.

We propose score function estimators for $k$-subset sampling (SFESS) as a *complement* to existing methods[1]. Our proposed approach fundamentally differs from prior works on $k$-subset sampling (see Table 1), offering both exact samples and unbiased gradient estimates. Furthermore, it does not assume differentiable downstream models, broadening the possible applications of $k$-subset sampling to cases when the downstream model's gradient is unavailable or computationally expensive.

In addition to the complementary advantages of our proposed approach, our research question holds significant relevance to the field, as previous work advises against the use of score function estimators for $k$-subset selection due to their high variance (Xie & Ermon, 2019; Niepert et al., 2021; Ahmed et al., 2023). Thus, illustrating the potential of this family of methods could facilitate further progress in a direction that is currently overlooked in the field.

To realize our proposal, we develop an efficient method for computing the score function based on the discrete Fourier transform (DFT) for computing the Poisson binomial distributions' PMF (Fernandez & Williams, 2010). Furthermore, we use control variates to significantly reduce the high variance of the vanilla score function estimator. In summary, our contributions are the following:

**Research gap** We identify and address a significant research gap in $k$-subset sampling where score function estimators are not being considered despite their conceptual simplicity, desirable properties, and broad applicability.

**Approach** We propose a score function estimator for the $k$-subset distribution featuring an efficient DFT-based score function calculation and reduced variance using multi-sample control variates.

**Results** We validate our approach in multiple experimental settings and find comparable results to state-of-the-art relaxed and approximate pathwise gradient methods, signifying the potential of score function estimators for $k$-subset selection.

## 2 PROBLEM STATEMENT AND MOTIVATION

**The gradient estimation problem** We are interested in learning with $k$-subset sampling using the following gradient:

$$\nabla_{\boldsymbol{\theta}} \mathbb{E}_{p_{\boldsymbol{\theta},k}(\boldsymbol{z})}[f(\boldsymbol{z})], \tag{1}$$

where $p_{\boldsymbol{\theta},k}$ is a parameterized distribution over subsets with size $k$ and $f$ is a downstream function of the subset samples. In practice, $f$ will often be a parameterized function with additional inputs besides $\boldsymbol{z}$. The discrete distribution over subsets is not amenable to the reparametrization trick (Kingma & Welling, 2014) which motivates the development of alternative gradient estimators for Eq. (1).

---

[1]Code available at https://github.com/klaswijk/sfess.

Table 1: **Method comparison**. Comparison of methods for learning with $k$-subset sampling based on the criteria: producing exact ($k$-hot) samples, having unbiased gradient estimates (a desirable property in statistical estimators), compatibility with non-differentiable objectives $f$, and being free from parameters requiring tuning (e.g., relaxation temperature, which may require multiple training runs to adjust). Insensitive parameters like the number of samples used for variance reduction are not considered tuned.

| Method | Exact samples | Unbiased | Allows non-differentiable $f$ | Tuning-free |
|---|:---:|:---:|:---:|:---:|
| GS (Xie & Ermon, 2019) | ✗ | ✓ | ✗ | ✗ |
| STGS (Xie & Ermon, 2019) | ✓ | ✗ | ✗ | ✗ |
| I-MLE (Niepert et al., 2021) | ✗ | ✗ | ✓ | ✗ |
| SIMPLE (Ahmed et al., 2023) | ✓ | ✗ | ✗ | ✓ |
| NCPSS (Pervez et al., 2023) | ✓[2] | ✗ | ✗ | ✓[4] |
| SFESS (Ours) | ✓ | ✓[3] | ✓ | ✓[4] |

**Existing approaches and their limitations**   Existing approaches to learning with $k$-subset sampling generally fall into one of two categories: approximate gradient methods or relaxed sampling.

Approximate pathwise gradient methods directly modify the gradient calculation. The best known example is the straight-through estimator (Bengio et al., 2013) which treats the sampling as an identity function during the backward pass. Recently, Liu et al. (2023) showed that straight-through estimation works as a first-order approximation of the gradient for Bernoulli and categorical samples. However, these approximate estimators tend to produce biased gradients.

Relaxed sampling methods replace the distribution with a relaxed distribution so that the reparametrization trick (Kingma & Welling, 2014) can be used to obtain a gradient. These are gradients of the relaxed samples, not the discrete samples of the original distribution. Regardless, these gradients can be used to train a model can be used with discrete samples at test time. Although this approach can often be effective, it has two significant limitations. One is that it requires using relaxed samples instead of discrete ones (which may not be possible depending on $f$), and the other is that there is a discrepancy between training and test time: the model trained with relaxed sampling is not guaranteed to generalize to discrete samples at test time. The error of this discrepancy is difficult to account for.

Common to both approximate pathwise gradients and relaxed sampling is the reliance on differentiable $f$, which limits their applicability to, e.g., non-differentiable settings in reinforcement learning or black-box functions. Figure 1 shows the forward and backward passes of the two approaches and how they differ from score function estimators. For in-depth reviews of Monte-Carlo gradient estimators and the Gumbel-max trick, we refer the reader to Mohamed et al. (2020) and Huijben et al. (2023) respectively.

**Why use black-box gradient estimates?**   A natural question to ask is what potential benefits black-box gradient estimates like score function estimators provide. Although discarding the pathwise gradient theoretically reduces the dimensionality of gradient information by one (Metz et al., 2021; Liu et al., 2023), it also allows for non-differentiable downstream functions. Interestingly, Metz et al. (2021) find that the variance of black-box estimates is not necessarily higher than for pathwise estimators. Furthermore, score function estimators have been used extensively in settings like variational inference (Ranganath et al., 2014) and reinforcement learning (Sutton et al., 1999) where they form the basis for algorithms like PPO (Schulman et al., 2017). We summarize some desirable properties of our proposed method, that stem from it being a score function estimator, and compare it to existing methods in Table 1. It is also worth noting that score function estimators have successfully been combined with pathwise estimators for Bernoulli and categorical distributions (Tucker et al., 2017; Grathwohl et al., 2018).

---

[2]NCPSS draws $k$-hot samples, but relaxes the subset size $k$ such that $k$ varies slightly.

[3]Using conditional Poisson samples in the forward pass.

[4]Ignoring the number of samples used for variance reduction, which only needs to be set sufficiently high.

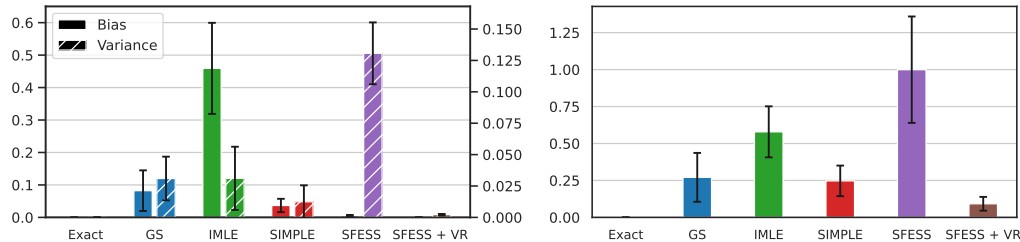

Figure 2: **Toy problem comparison**. Bias and variance (left) and error (right) of gradient estimates in a toy problem (Ahmed et al., 2023) with known ground-truth gradients. All methods use single sample estimates, except SFESS + VR, where control variates are computed using 32 samples. Estimates are computed using 10,000 samples, with error bars (1 std) from 10 repetitions with different random seeds.

## 3 METHOD

We are interested in devising a black-box gradient estimator for $k$-subset sampling with efficacy similar to the existing techniques. Here, we describe our method including how to compute the score function and reduce its variance with control variables. The resulting algorithm is presented in Algorithm 2 along with Gumbel top-$k$ sampling (Kool et al., 2019b) in Algorithm 1 for $k$-subset sampling.

**Overview** We are interested in sampling subsets $\boldsymbol{z}$ of size $k$ given a set of $n$ variables. We consider the following distribution:

$$p_{\boldsymbol{\theta},k}(\boldsymbol{z}) = p_{\boldsymbol{\theta}}\left(\boldsymbol{b} \mid \sum_{i=1}^{n} b_i = k\right) = \frac{\prod_{i=1}^{n} p_{\boldsymbol{\theta}}(b_i)}{p_{\boldsymbol{\theta}}\left(\sum_{i=1}^{n} b_i = k\right)} \mathbf{1} \sum_{i=1}^{n} b_i = k, \quad (2)$$

where $\boldsymbol{b} \in \{0,1\}^n$ is independently Bernoulli distributed withf parameters $\boldsymbol{\theta} \in [0,1]^n$ and $\mathbf{1}$ denotes the indicator function. This equation induces a particular distribution over the $\binom{n}{k}$ possible subsets using only $n$ parameters and is naturally only one of many ways to do so. In sampling design, this particular approach is known as conditional Poisson sampling (Tillé, 2006).

Previous work has explored approximate pathwise derivatives of various $k$-subset distributions' samples (Xie & Ermon, 2019; Ahmed et al., 2023). In this work, we instead consider score function estimators that are *exact* in expectation. Hence, we want to compute the score function defined on the region where $\sum_{i=1}^{n} b_i = k$,

$$\nabla_{\boldsymbol{\theta}} \log p_{\boldsymbol{\theta},k}(\boldsymbol{z}) = \sum_{i=1}^{n} \underbrace{\nabla_{\boldsymbol{\theta}} \log p_{\boldsymbol{\theta}}(b_i)}_{\text{Bernoulli}} - \underbrace{\nabla_{\boldsymbol{\theta}} \log p_{\boldsymbol{\theta}}\left(\sum_{i=1}^{n} b_i = k\right)}_{\text{Poisson binomial}}. \quad (3)$$

Computing the first term is easy, since each $p_{\boldsymbol{\theta}}(b_i)$ is Bernoulli distributed. The second term appears more challenging. It is the score function of a Poisson binomial distribution, a generalized binomial distribution where the samples are not necessarily identically distributed. Several efficient methods for computing the Poisson binomial's PMF have been proposed, including approximate and recursive methods (Le Cam, 1960; Wadycki et al., 1973; Ahmed et al., 2023). We follow Fernandez & Williams (2010) and compute it using an FFT (Cooley & Tukey, 1965), leveraging its $\mathcal{O}(n \log n)$ time-complexity and efficient implementation on modern hardware[5]. The gradient of the log probability is computed using automatic differentiation.

Now, being able to compute the score function in Eq. (3), we can write the following score function estimator:

$$\nabla_{\boldsymbol{\theta}} \mathbb{E}_{p_{\boldsymbol{\theta},k}(\boldsymbol{z})}[f(\boldsymbol{z})] = \mathbb{E}_{p_{\boldsymbol{\theta},k}(\boldsymbol{z})}[\nabla_{\boldsymbol{\theta}} \log p_{\boldsymbol{\theta},k}(\boldsymbol{z}) f(\boldsymbol{z})] \approx \frac{1}{N} \sum_{i=1}^{N} \nabla_{\boldsymbol{\theta}} \log p_{\boldsymbol{\theta},k}(\boldsymbol{z}^{(j)}) f(\boldsymbol{z}^{(i)}), \quad (4)$$

where $N$ is the number of $k$-subset samples $\boldsymbol{z}^{(j)} \sim p_{\boldsymbol{\theta},k}(\boldsymbol{z})$ used in the Monte-Carlo estimate of the expectation. For completeness, we derive the standard score function estimator in Appendix A.

---

[5]We use the Nvidia cuFFT implementation in PyTorch. See Appendix B for pseudocode.

---

**Algorithm 1** Subset sampling using Gumbel top-$k$

---

**Require:** Subset parameters $\boldsymbol{\theta}$ and size $k$

1: Sample noise $g_i \sim \text{Gumbel}(0, 1)$ for $i = 1, \ldots, n$

2: Compute $\boldsymbol{z} \leftarrow \text{ArgTopK}(\log \boldsymbol{\theta} + \boldsymbol{g}, \; k)$       $\triangleright$ A $k$-hot vector

3: **return** $\boldsymbol{z}$

---

**Algorithm 2** SFESS + VR: Score function estimator for $k$-subset sampling with variance reduction

---

**Require:** Initial subset parameters $\boldsymbol{\theta}$ and size $k$, and number of variance reduction samples $N$

1: **repeat**

2:   Sample $\boldsymbol{z}^{(i)} \sim p_{\boldsymbol{\theta},k}(\boldsymbol{z})$ for $i = 1, \ldots, N$     $\triangleright$ Or conditionally with, e.g., $p_{\boldsymbol{\theta},k}(\boldsymbol{z}|\boldsymbol{x})$

3:   Compute the Poisson-Binomial likelihood $\log p_{\boldsymbol{\theta}}\left(\sum_{i=1}^{n} b_i^{(j)} = k\right)$ using Eq. (6)

4:   Compute the score function $\nabla_{\boldsymbol{\theta}} \log p_{\boldsymbol{\theta},k}(\boldsymbol{z}^{(i)})$ using Eq. (3) and autodiff

5:   Evaluate $f(\boldsymbol{z}^{(i)})$ for $i = 1, \ldots, N$      $\triangleright$ Or with additional inputs, e.g., $f(\boldsymbol{z}, \boldsymbol{x})$

6:   Optimize parameters $\boldsymbol{\theta}$ using the variance-reduced gradients in Eq. (7)

7: **until** convergence            $\triangleright$ Number of steps, threshold, etc.

8: **return** $\boldsymbol{\theta}$

---

**Efficiently computing the score function**   The second term of Eq. (3) follows a Poisson binomial distribution. The likelihood of which can be written as:

$$p_{\boldsymbol{\theta}}\left(\textstyle\sum_{i=1}^{n} b_i = k\right) = \sum_{\boldsymbol{b} \in \{0,1\}^n} p_{\boldsymbol{\theta}}(\boldsymbol{b}) \mathbf{1}_{\sum_{i=1}^{n} b_i = k}. \tag{5}$$

Computing the PMF using Eq. (5) requires iterating all $2^n$ binary vectors $\boldsymbol{b}$ which is prohibitively expensive. Instead, we look for a more efficient method. Eq. (5) in Fernandez & Williams (2010) gives us this closed-form expression:

$$p_{\boldsymbol{\theta}}\left(\textstyle\sum_{i=1}^{n} b_i = k\right) = \frac{1}{n+1} \sum_{l=0}^{n} \left( e^{-lk\frac{2\pi i}{n+1}} \prod_{m=1}^{n} \left[ p_{\boldsymbol{\theta}}(b_m) e^{l\frac{2\pi i}{n+1}} + (1 - p_{\boldsymbol{\theta}}(b_m)) \right] \right), \tag{6}$$

where $i = \sqrt{-1}$. The corresponding discrete Fourier transform (Eq. (6) in Fernandez & Williams (2010)):

$$\text{DFT}\left( \prod_{m=1}^{n} \left[ p_{\boldsymbol{\theta}}(b_m) e^{-lk\frac{2\pi i}{n+1}} + (1 - p_{\boldsymbol{\theta}}(b_m)) \right] \right) \quad l = 0, \ldots, n,$$

is efficiently solved for all $k$ using an FFT Cooley & Tukey (1965). Note that the PMF is a function of $\boldsymbol{\theta}$ and $k$ so it does not need to be recomputed when evaluating Eq. (3) for different samples $\boldsymbol{z}$. This is useful for computing the control variate in Eq. (7) where we draw multiple samples with the same $\boldsymbol{\theta}$.

**Reducing variance with control variates**   The vanilla score function estimator generally suffers from high variance. While many variance reduction techniques have been proposed (Mnih & Gregor, 2014; Gu et al., 2016; Tucker et al., 2017; Shi et al., 2022), we choose to employ control variates using multiple samples (Mnih & Rezende, 2016; Kool et al., 2019a) in this work due to its simplicity, unbiasedness, and lack of additional assumptions. In Section 4, we will see that this straightforward approach proves highly effective. The estimator with reduced variance is shown below:

$$\nabla_{\boldsymbol{\theta}} \mathbb{E}_{p_{\boldsymbol{\theta},k}(\boldsymbol{z})}[f(\boldsymbol{z})] \approx \frac{1}{N} \sum_{i=1}^{N} \nabla_{\boldsymbol{\theta}} \log p_{\boldsymbol{\theta},k}(\boldsymbol{z}^{(j)}) \left( f(\boldsymbol{z}^{(j)}) - \frac{1}{N-1} \sum_{i \neq j} f(\boldsymbol{z}^{(j)}) \right). \tag{7}$$

**Drawing approximate samples**   Having designed our gradient estimator for samples following the distribution in Eq. (2), we now turn to the sampling operation in the forward pass. Sampling from

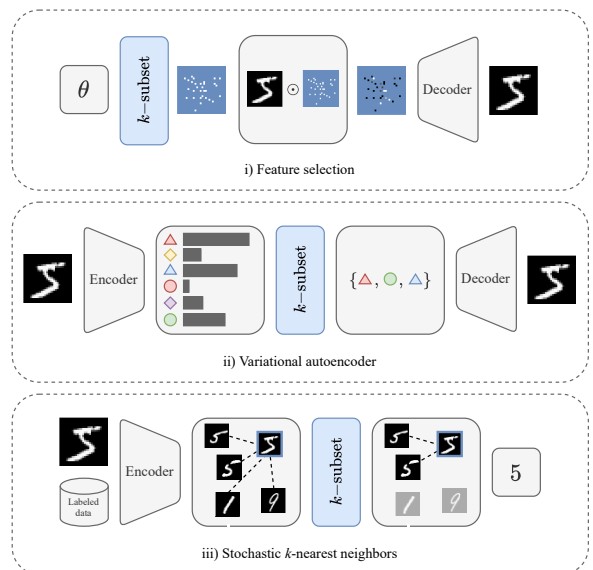

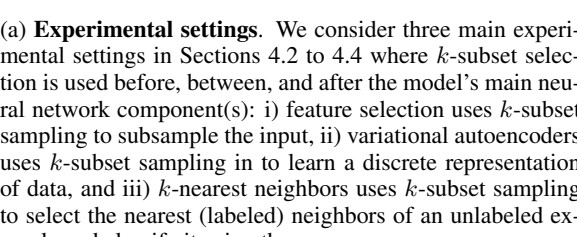

(a) **Experimental settings**. We consider three main experimental settings in Sections 4.2 to 4.4 where $k$-subset selection is used before, between, and after the model's main neural network component(s): i) feature selection uses $k$-subset sampling to subsample the input, ii) variational autoencoders uses $k$-subset sampling in to learn a discrete representation of data, and iii) $k$-nearest neighbors uses $k$-subset sampling to select the nearest (labeled) neighbors of an unlabeled example and classify it using them.

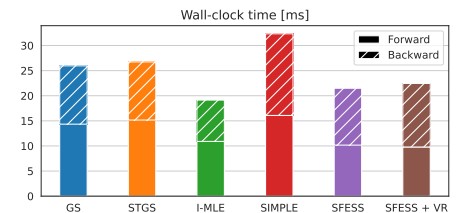

(b) **Wall-clock time**. The average wall clock time of a single training step of the methods in Table 3. Despite drawing 32 samples for variance reduction, the increase in wall time from SFESS to SFESS + VR is minor.

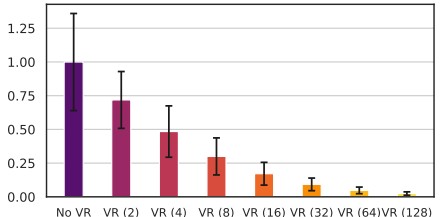

(c) **Variance reduction and gradient error**. The cosine difference of the true gradient and the estimated gradient using SFESS + VR with different numbers of variance reduction samples on the toy problem with known gradients. No VR corresponds to the vanilla SFESS estimator. Estimates are computed using 10,000 samples, with error bars (1 std) from 10 repetitions with different random seeds.

the distribution described by Eq. (2), i.e., conditional Poisson sampling, presents two challenges. First, sampling procedures for generating conditional Poisson samples assume parameters $\boldsymbol{\theta}$ such that $\sum_{i=1}^{n} \theta_i = k$. Second, drawing samples efficiently is more challenging than, e.g., Gumbel top-$k$ sampling, with most procedures using rejection sampling. In practice, we find that bypassing these problems by using Gumbel-top-$k$ samples as approximate conditional Poisson samples gives satisfactory results and makes for a simple and efficient forward pass.

**Conditional distributions and variants of $f$** Conditional $k$-subset distributions $p_{\boldsymbol{\theta}}(\boldsymbol{z}|\boldsymbol{x})$ are a useful extension of the model presented above that do not change the gradient estimator (the estimated gradients are simply backpropagated through the conditioning variable). Similarly, downstream functions with additional inputs, e.g., $f(\boldsymbol{z}, \boldsymbol{x})$, and parameterized functions, e.g., $f_{\phi}$, are easily incorporated and optimized alongside the $k$-subset distribution's parameters. We investigate both conditional distribution and neural-network parameterized functions in our experiments (Section 4).

## 4 EXPERIMENTS

In this section, we validate our proposed estimator in three main experimental settings: feature selection, variational autoencoders (VAE), and stochastic $k$-nearest-neighbors ($k$-NN). In this set of problems, the $k$-subset distribution is used in various ways: as the first operation in feature selection, as the mid-point bottleneck in a VAE, and in computing the final loss in stochastic $k$-NN (see Figure 3a).

We use MNIST (LeCun et al., 1998) and FASHION MNIST (Xiao et al., 2017) with the canonical train and test splits. We withhold 10,000 samples from the train set for validation. For all training, we use a batch size of 128 and train for 50,000 steps using the Adam optimizer (Kingma & Ba, 2015) with a learning rate of 1e−4 and parameters $\beta_1 = 0.9$ and $\beta_2 = 0.999$. We compare our proposed method with variance reduction (SFESS + VR) using 32 variance reduction samples to

relaxed subset sampling (GS) and its straight-through variant (STGS) (Xie & Ermon, 2019), implicit maximum likelihood estimation (I-MLE) (Niepert et al., 2021), SIMPLE (Ahmed et al., 2023), and SFESS without variance reduction. For ST and STGS we use the the relaxation temperature $\tau = 0.5$, which gave the best overall results out of $\tau \in \{0.1, 0.5, 1.0\}$. For I-MLE, we set both the input and target noise temperature to 1.0. As noted in Table 1, SIMPLE and our method have no hyperparameters in need of tuning.

## 4.1 Toy Problem

First, we consider a simple toy setting with known ground-truth gradients. We adapt the toy problem in Ahmed et al. (2023)[6] where the gradient estimator is used to minimize $\mathbb{E}_{p_{\boldsymbol{\theta}}(\boldsymbol{z})}[\|\boldsymbol{z} - \boldsymbol{\theta}^*\|^2]$ where $\boldsymbol{\theta}^*$ are the ground-truth parameters sampled from a standard normal distribution. Using $n = 10$ and $k = 5$ lets us enumerate all $\binom{10}{5} = 256$ subsets and compute the ground-truth gradient. Figure 2 shows the estimated bias, variance, and error ($1 - \text{cosine similarity compared to ground-truth}$) of the different estimators. Figure 3c shows the decreasing error of SFESS + VR as the number of variance reduction samples increases.

## 4.2 Feature selection

Sampling a subset of inputs and estimating the gradients (Balın et al., 2019; Huijben et al., 2019; Yamada et al., 2020) is an intuitive approach to differentiable feature selection. By being differentiable, the selection can be jointly optimized alongside a downstream network. We consider feature selection for reconstruction and where a reconstruction network ($28^2 \to 200 \to 200 \to 28^2$ dense ReLU network) predicts the full set of input features inputs masked by the sampled subset and both the subset parameters and reconstruction network are optimized using the reconstruction loss (binary cross entropy). Table 2 shows our results and Figure 4 the convergence of the validation loss.

## 4.3 Variational Autoencoders

Variational Autoencoders (Kingma & Welling, 2014) with latent variables distributed over $k$-subsets has been used as a benchmark in previous work on learning with $k$-subset sampling (Niepert et al., 2021; Ahmed et al., 2023). We use the approximate ELBO and network architecture of Niepert et al. (2021). The encoder ($28^2 \to 512 \to 256 \to nd$ dense ReLU network) encodes the input. The outputs are reshaped to ($d \times d$). Then, $d$ $k$-subset sample of length $n$ are drawn and decoded by the decoder ($d^2 \to 256 \to 512 \to 28^2$ dense ReLU network). The loss is the sum of a reconstruction loss (binary cross entropy) and the KL-divergence between each latent distribution and a uniform prior. Table 3 shows our results and Figure 5 the convergence of the validation loss. Finally, the wall-clock time is shown in Figure 3b.

## 4.4 Stochastic $k$-Nearest-Neighbors

Our final experiment is stochastic $k$-NN (Grover et al., 2019). Here, we learn an embedding that optimizes the classification accuracy of $k$-NN. During training, we sample a query point $\{\boldsymbol{x}_{\text{query}}, \boldsymbol{y}_{\text{query}}\}$ and a batch of neighbors $\{\boldsymbol{x}_{\text{neighbor}}^{(i)}, \boldsymbol{y}_{\text{neighbor}}^{(i)}\}_{i=1}^{n}$ (we use $n = 128$ in our experiments) and encode them using an encoder $g_{\boldsymbol{\phi}}$ ($28^2 \to 512 \to 256 \to d$ dense ReLU network). Then, we compute the Euclidean distances from the query point embedding to all neighbor candidates' embeddings $\{\|g_{\boldsymbol{\phi}}(\boldsymbol{x}_{\text{query}}) - g_{\boldsymbol{\phi}}(\boldsymbol{x}_{\text{neighbor}}^{(i)})\|\}_{i=1}^{n}$ and sample a $k$-subset of neighbors using the distances as unnormalized logits. Finally, the negated proportion of the $k$-subset with the same label as the query point is used as a loss. The algorithm is slightly different at test time: we use the entire training set as candidate neighbors and compute the $k$-nearest-neighbors deterministically instead of sampling a $k$-subset. Table 4 shows the results. The convergence of accuracy on the validation set is shown in Appendix C. Embeddings of the validation sets are shown in Figure 6.

---

[6]Code available at `https://github.com/UCLA-StarAI/SIMPLE`.

Table 2: **Feature selection results**. BCE on the test split. The parameters $n$ and $k$ are the number of inputs and the number of selections respectively. The means and standard deviations are computed from 5 repetitions with different random seeds. The best mean result is shown in **bold** and the second best mean result is underlined.

| Method | $n$ | $k$ | MNIST | | FASHION MNIST | |
|---|---|---|---|---|---|---|
| | | | Mean | Std | Mean | Std |
| GS (Xie & Ermon, 2019) | 784 | 50 | 0.147 | ± 0.005 | 0.320 | ± 0.002 |
| STGS (Xie & Ermon, 2019) | 784 | 50 | 0.146 | ± 0.001 | 0.318 | ± 0.002 |
| I-MLE (Niepert et al., 2021) | 784 | 50 | 0.182 | ± 0.010 | 0.323 | ± 0.001 |
| SIMPLE (Ahmed et al., 2023) | 784 | 50 | 0.133 | ± 0.001 | 0.311 | ± 0.001 |
| SFESS (Ours) | 784 | 50 | 0.189 | ± 0.011 | 0.326 | ± 0.002 |
| SFESS + VR (Ours) | 784 | 50 | **0.132** | ± 0.002 | **0.307** | ± 0.001 |
| GS (Xie & Ermon, 2019) | 784 | 30 | 0.168 | ± 0.004 | 0.336 | ± 0.002 |
| STGS (Xie & Ermon, 2019) | 784 | 30 | 0.173 | ± 0.005 | 0.335 | ± 0.004 |
| I-MLE (Niepert et al., 2021) | 784 | 30 | 0.206 | ± 0.010 | 0.341 | ± 0.005 |
| SIMPLE (Ahmed et al., 2023) | 784 | 30 | 0.160 | ± 0.002 | 0.327 | ± 0.002 |
| SFESS (Ours) | 784 | 30 | 0.214 | ± 0.011 | 0.343 | ± 0.004 |
| SFESS + VR (Ours) | 784 | 30 | **0.154** | ± 0.003 | **0.320** | ± 0.002 |

Table 3: **VAE results**. BCE + KL-divergence on the test set. The parameters $d$, $n$, and $k$ are the number of latent subsets, their dimensionality, and size respectively. The means and standard deviations are computed from 5 repetitions with different random seeds. The best mean result is shown in **bold** and the second best mean result is underlined.

| Method | $d$ | $n$ | $k$ | MNIST | | FASHION MNIST | |
|---|---|---|---|---|---|---|---|
| | | | | Mean | Std | Mean | Std |
| GS (Xie & Ermon, 2019) | 10 | 10 | 5 | 97.36 | ± 2.08 | 241.72 | ± 1.57 |
| STGS (Xie & Ermon, 2019) | 10 | 10 | 5 | 95.05 | ± 1.57 | 233.68 | ± 0.53 |
| I-MLE (Niepert et al., 2021) | 10 | 10 | 5 | 99.74 | ± 0.77 | 234.88 | ± 0.36 |
| SIMPLE (Ahmed et al., 2023) | 10 | 10 | 5 | **81.90** | ± 0.10 | **225.19** | ± 0.11 |
| SFESS (Ours) | 10 | 10 | 5 | 205.72 | ± 0.15 | 384.27 | ± 1.20 |
| SFESS + VR (Ours) | 10 | 10 | 5 | 90.04 | ± 2.79 | 227.73 | ± 0.12 |
| GS (Xie & Ermon, 2019) | 20 | 20 | 10 | 86.25 | ± 1.03 | 248.63 | ± 1.87 |
| STGS (Xie & Ermon, 2019) | 20 | 20 | 10 | 73.90 | ± 0.24 | 225.06 | ± 0.55 |
| I-MLE (Niepert et al., 2021) | 20 | 20 | 10 | 84.55 | ± 0.45 | 238.13 | ± 1.95 |
| SIMPLE (Ahmed et al., 2023) | 20 | 20 | 10 | **67.96** | ± 0.14 | 218.82 | ± 0.29 |
| SFESS (Ours) | 20 | 20 | 10 | 205.86 | ± 0.05 | 384.81 | ± 0.11 |
| SFESS + VR (Ours) | 20 | 20 | 10 | 68.83 | ± 0.15 | **218.39** | ± 0.15 |

Table 4: $k$-**NN results**. Accuracy on the test set. The parameters $d$, $n$, and $k$ are the dimensionality of the embedding, the number of neighbors sampled in the training steps, and the parameter of $k$-NN respectively. The means and standard deviations are computed from 5 repetitions with different random seeds. The best mean result is shown in **bold** and the second best mean result is underlined.

| Method | $d$ | $n$ | $k$ | MNIST | | FASHION MNIST | |
|---|---|---|---|---|---|---|---|
| | | | | Mean | Std | Mean | Std |
| GS (Xie & Ermon, 2019) | 2 | 128 | 10 | **0.950** | ± 0.002 | **0.873** | ± 0.002 |
| STGS (Xie & Ermon, 2019) | 2 | 128 | 10 | **0.950** | ± 0.002 | **0.873** | ± 0.002 |
| I-MLE (Niepert et al., 2021) | 2 | 128 | 10 | 0.740 | ± 0.037 | 0.696 | ± 0.023 |
| SIMPLE (Ahmed et al., 2023) | 2 | 128 | 10 | 0.949 | ± 0.002 | 0.871 | ± 0.002 |
| SFESS (Ours) | 2 | 128 | 10 | 0.938 | ± 0.009 | 0.778 | ± 0.010 |
| SFESS + VR (Ours) | 2 | 128 | 10 | 0.949 | ± 0.002 | 0.869 | ± 0.001 |

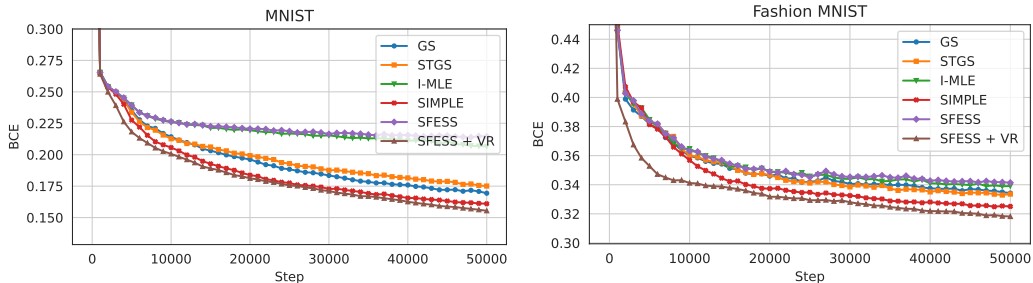

Figure 4: **Feature selection validation loss**. Convergence of BCE on the validation set for feature selection with $k = 30$ selections (see Appendix C for $k = 50$) averaged over 5 repetitions with different random seeds. The results follow the trend in the toy experiment (see Figure 2).

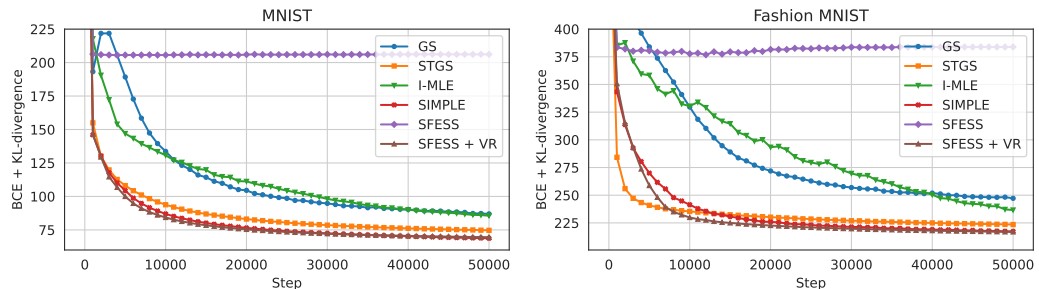

Figure 5: **VAE validation loss**. Convergence of BCE + KL-divergence on the validation set with $d = 20$, $n = 20$, and $k = 10$ (see Appendix C for $d = 10$, $n = 10$, and $k = 5$) averaged over 5 repetitions with different random seeds. The effect of variance reduction on SFESS is evident—going from a failure to learn useful representations to second best among the methods tested.

### 4.5 ADDITIONAL RESULTS

We provide a few additional studies shedding light on the various empirical aspects of SFESS. First, we explore the benefit of variance reduction. Figure 3c shows the improved alignment of the gradients with ground truth on the toy dataset. This improvement comes at the cost of increased sampling which could affect the computational burden of SFESS. Figure 3b shows the wall-clock time of SFESS and other baselines, indicating only a minor increase in the total wall-clock time due to the additional sampling. Finally, while the quantitative evaluation metrics indicate the efficacy of SFESS+VR compared to baselines, in Figure 6 we illustrate the quality of learned embedding.

## 5 RELATED WORK

In this section, we provide an overview of existing methods for $k$-subset sampling. Table 1 shows a qualitative comparison of the methods' different properties.

**Relaxed Subset Sampling** (Xie & Ermon, 2019) extends the Gumbel-Softmax distribution to distributions over subsets. Despite its elegance, relaxed subset sampling inherits the biased gradient estimation of the Gumbel-Softmax estimator. Furthermore, the top-$k$ sampling procedure sequentially applies the softmax function $k$ times, which limits scalability with respect to $k$ and potentially degrades performance (Pervez et al., 2023). The temperature parameter $\tau \in \mathbb{R}_{\geq 0}$ controls the relaxation strength. The relaxed samples approach uniform as $\tau \to \infty$ and $k$-hot as $\tau \to 0$.

**SIMPLE** (Ahmed et al., 2023) approximates the pathwise gradient of the sample using its exact marginals, achieving both lower bias and variance than ST Gumbel-Softmax top-$k$.

**Neural Conditional Poisson Subset Sampling (NCPSS)** (Pervez et al., 2023) relaxes $k$-subset sampling in a manner different from relaxed subset sampling (Xie & Ermon, 2019), allowing subset sizes slightly smaller and larger subsets than $k$. Then, pathwise gradient estimates are used for

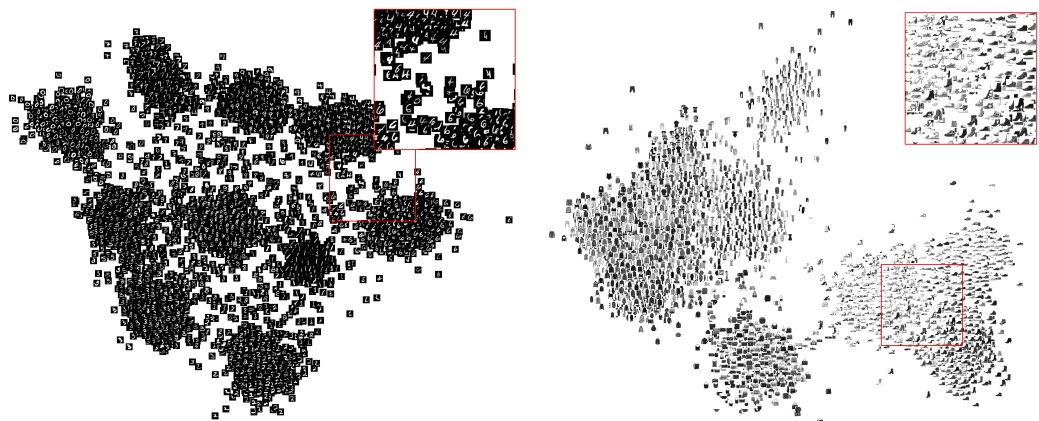

Figure 6: **Stochastic $k$-NN embeddings**. Two-dimensional embeddings ($d = 2$) of the MNIST (left) and FASHION MNIST (right) validation sets learned by optimizing the stochastic $k$-NN objective with $k = 10$ for 30,000 training steps. The resulting embeddings form clusters of the same class. Note that some of the samples placed between clusters are indeed ambiguous examples.

differentiable optimization. The authors show that NCPSS is more scalable than relaxed subset sampling and that the subset size $k$ can be optimized alongside the distribution's parameters.

**Implicit Maximum Likelihood Estimation (I-MLE)** (Niepert et al., 2021) uses a perturb-and-MAP approach that is applicable to general optimization problems, with subset sampling as a special case.

**Other methods** In some settings, a subset distribution can be modeled as either the concatenation of $n$ Bernoulli variables or the sum of $k$ categorical variables. This way, a host of gradient estimates for Bernoulli and categorical variables can be used (Yamada et al., 2020; Paulus et al., 2021; Dimitriev & Zhou, 2021; Shi et al., 2022; De Smet et al., 2023; Liu et al., 2023). However, neither option directly models $k$-subset sampling. Bernoulli variables require some constraint (e.g., a loss term) limiting the subset size, and a sum of categoricals requires $nk$ parameters and runs the risk of duplicate inclusions (Nilsson et al., 2024). Finally, there are techniques for relaxed sampling of other discrete structures like permutation matrices, trees, or graphs (Paulus et al., 2020).

## 6 CONCLUSION

In this work, we identified a research gap to explore the viability of score-function estimators for learning with $k$-subset sampling. We devised a simple approach and showed its efficacy in a variety of tasks achieving comparable results to existing state-of-the-art. This is a significant finding not only due to the complementary properties and wider applicability of our approach but also due to its dismissal in the current literature. The main limitation of our proposed estimator is the need to draw multiple samples for variance reduction. This means we need to evaluate $f$ multiple times, which can be costly or impossible in some settings. Another limitation is using Gumbel top-$k$ samples as approximate conditional Poisson samples in the forward pass, which introduces some bias in practice. We believe our work paves the way for future research in differentiable $k$-subset sampling, such as combining score-function and pathwise estimators similarly to what was done for Bernoulli and categorical distributions. (Tucker et al., 2017; Grathwohl et al., 2018).

ACKNOWLEDGMENTS

The authors would like to thank Denis Korzhenkov for bringing our attention to the differences between Gumbel top-$k$ sampling and conditional Poisson sampling. We would also like to thank Luca Franceschi for valuable feedback and for connecting us with Andrei-Marian Manolache, Ahmed Kareem, and Mathias Niepert who assisted greatly with applying SIMPLE.

This work was partially supported by the Swedish e-Science Research Centre (SeRC) and KTH Digital Futures. The computations were enabled by resources provided by the National Academic Infrastructure for Supercomputing in Sweden (NAISS), partially funded by the Swedish Research Council through grant agreement no. 2022-06725 as well as the Berzelius resource provided by the Knut and Alice Wallenberg Foundation at the National Supercomputer Centre.

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

# A    DERIVING THE SCORE FUNCTION ESTIMATOR

In this appendix, we derive the score function estimator (Williams, 1992) which provides a Monte-Carlo estimate of the objective's gradient. We adapt the proof from Mohamed et al. (2020) (with annotations added):

$$\nabla_{\boldsymbol{\theta}}\mathbb{E}_{p_{\boldsymbol{\theta}}(\boldsymbol{z})}[f(\boldsymbol{z})] = \nabla_{\boldsymbol{\theta}}\sum_{\boldsymbol{z}} p_{\boldsymbol{\theta}}(\boldsymbol{z})f(\boldsymbol{z}) \qquad \text{By definition of } \mathbb{E} \qquad (8)$$

$$= \sum_{\boldsymbol{z}} \nabla_{\boldsymbol{\theta}} p_{\boldsymbol{\theta}}(\boldsymbol{z})f(\boldsymbol{z}) \qquad \text{Interchange gradient and summation}$$

$$= \sum_{\boldsymbol{z}} p_{\boldsymbol{\theta}}(\boldsymbol{z})\nabla_{\boldsymbol{\theta}} \log p_{\boldsymbol{\theta}}(\boldsymbol{z})f(\boldsymbol{z}) \qquad \text{By log derivative rule}$$

$$= \mathbb{E}_{p_{\boldsymbol{\theta}}(\boldsymbol{z})}[f(\boldsymbol{z})\nabla_{\boldsymbol{\theta}} \log p_{\boldsymbol{\theta}}(\boldsymbol{z})] \qquad \text{By definition of } \mathbb{E} \qquad (9)$$

$$\approx \frac{1}{N}\sum_{i=1}^{N} f(\boldsymbol{z}^{(i)})\nabla_{\boldsymbol{\theta}} \log p_{\boldsymbol{\theta}}(\boldsymbol{z}^{(i)}) \qquad \text{Monte-Carlo estimate} \qquad (10)$$

By the law of large numbers, the Monte-Carlo estimator in Eq. (10) converges to the expected value in Eq. (9) as $N \to \infty$, which is exactly the value of the true gradient in Eq. (8). Hence, the estimator is an unbiased estimator of the true gradient.

# B    SCORE FUNCTION CALCULATION

A key component of SFESS is calculating the score function in Eq. (3). The unconditional independent Bernoulli distribution is renormalized by the Poisson binomial distribution. This renormalization factor is calculated following Fernandez & Williams (2010). Listing 1 outlines this calculation in pseudocode.

**Listing 1** PyTorch-style pseudocode for calculating the Poisson-Binomial PMF (Fernandez & Williams, 2010).

```python
import torch
import cmath

def poibin_prob(theta, k):
    n = theta.size(0)
    i = torch.arange(n + 1).unsqueeze(-1)
    c = cmath.exp(2j * torch.pi / (n + 1))
    prod = torch.prod(theta * c**i + (1 - theta), dim=1)
    probs = torch.fft.fft(prod).real / (n + 1)
    return probs[k]
```

# C ADDITIONAL LOSS CURVES

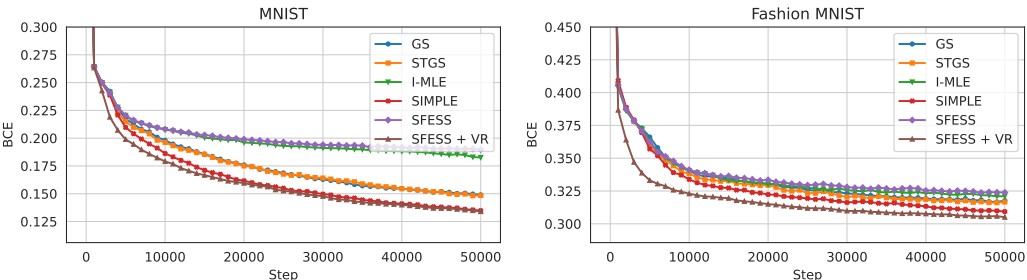

Figure 7: **Feature selection validation loss**. Convergence of BCE on the validation set for feature selection with for $k = 50$ averaged over 5 repetitions with different random seeds.

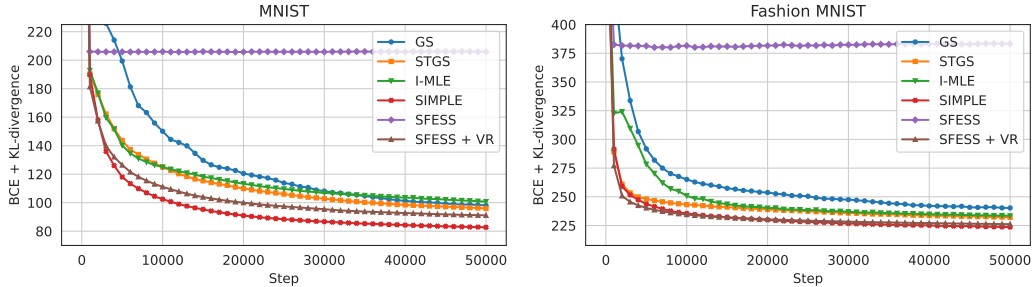

Figure 8: **VAE validation loss**. Convergence of BCE + KL-divergence on the validation set with $d = 10$, $n = 10$, and $k = 5$ averaged over 5 repetitions with different random seeds.

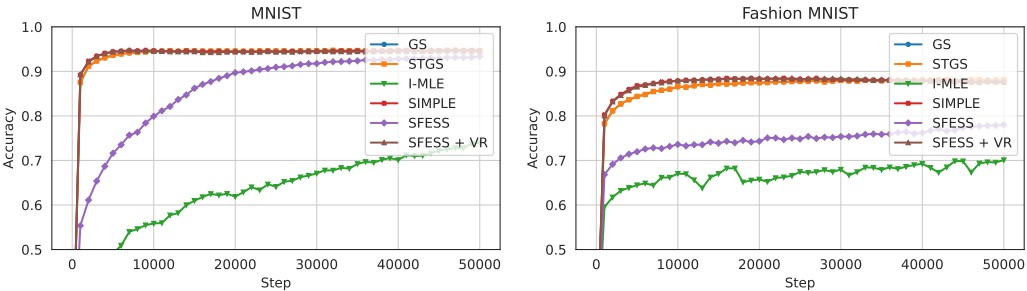

Figure 9: $k$-**NN validation accuracy**. Convergence of accuracy on the validation set with $d = 2$, $n = 128$, and $k = 10$ averaged over 5 repetitions with different random seeds.

