# OpenReview forum: "SFESS: Score Function Estimators for $k$-Subset Sampling"
_ICLR.cc/2025/Conference — ICLR 2025 Poster_

### Official Review · Reviewer_6UyM · 2024-10-25

**Soundness:** 3
**Presentation:** 2
**Contribution:** 2
**Rating:** 6
**Confidence:** 2

**Summary:**

This paper studies the k-subset sampling and proposes a differentiable estimator that enjoys the benefits of score function estimators and the relaxed approperiate pathwise estimator. The proposed method extends the existing differentiable optimization of Bernoulli and categorical distributions and serves as a complement to existing methods.

**Strengths:**

I am not very familiar with the topic but I believe this work has made significant contributions. In traditional k-subset sampling methods, the score function estimators are rarely considered. This paper fills this gap by proposing the proposed method. The results are validated on diverse experiments and match the SOTA relaxed and approximate pathwise gradient methods. Therefore, the method is original and significantly improves the existing approaches.

**Weaknesses:**

The backgrounds and introduction to the studied topic are not very friendly. It is hard for me to understand why the k-subset sampling is needed and why the existing methods (which may not be based on the score function and potentially non-differentiable) are not sufficient to the related community. I hope the author could add more examples, explicitly stating where the k-subset sampling is used and why the existing methods fail to satisfy the demond.

**Questions:**

In Table 1: Method Comparison, the author lists four properties (Exact Samples, ...). Can the author explain why these properties are important or preferred to have? What will happen, for example, if the estimator is non-differentiable?

When we need to have the gradient of some methods such as GS, STGS, or SIMPLE, can I simply use a gradient estimation approach to approximate their gradient? Does it make any difference from this method?

A minor issue:

I probably disagree that the variance reduction using some control variates can be considered as any contribution, since it has been well-known in the literature. However, if the VR is not employed in the SFESS method, its performance in Figure 5, is nearly the worst among all methods. How should I understand this figure?

---

> ### Author Response · Authors · 2024-11-21
> **Author Response to Reviewer 6UyM**
>
> Thank you for your valuable feedback. We address your comments and questions below.
>
> **W1: Where is $k$-subset sampling used?**
>
> The first paragraph of the introduction lists some examples of where differentiable $k$-subset sampling can be used. Generally, it is used to include $k$-hot vectors in differentiable optimization by sampling the vector and estimating its gradient. There are three examples in the experiment section: choosing multiple options at once (feature selection), learning a sparse code (VAE), and a top-$k$ operation (stochastic k-NN). The main limitations of existing methods are their biased gradients and reliance on the downstream function’s gradients (see the paragraph starting on line 136 and Table 1). As for applications, $k$-subset sampling has broad uses, e.g. MRI reconstruction [1], microarray data [2], and explainability [3].
>
> **Q1: Importance of properties in Table 1**
>
> Here we clarify why the properties in Table 1 are important and preferred:
> - *Exact samples.* This is preferred for two reasons (also mentioned on line 146): (1) it makes the downstream model consistent at training and test time (facilitating generalization), and (2) it allows for training with downstream functions that only accept discrete inputs and cannot be relaxed. Consider, for instance, a reinforcement learning setting where the action space is a subset (e.g. press $k$ out of $n$ buttons). In fact, obtaining exact samples was a motivating idea behind SIMPLE.
> - *Unbiased.* An unbiased estimator’s expected value is equal to the true parameter. This is generally a desirable property in statistical estimators and particularly important when analyzing convergence (see W1 in our response to Reviewer W34X).
> - *Non-differentiable.* This indicates whether the gradient estimate can be computed for a non-differentiable downstream function $f$ (see the table caption). The column header is now changed to “Non-differentiable $f$” in the revised paper to improve clarity. See W2 in our response to Reviewer kBPo for particular use cases of this property.
> - *Tuning-free.* This property is desirable because it (1) simplifies the model and (2) reduces the computational burden of hyperparameter tuning. In GS, STGS, and IMLE, tuning the temperature parameter for a given problem is crucial, often requiring repeated training runs.
>
> We have revised the caption of Table 1 to clarify points that were not already in the paper.
>
> **Q2: Estimating gradients**
>
> We are unsure about the intended question, so, we considered two possible interpretations below and answered them accordingly. Let us know if they do not answer your question.
>
> *Interpretation 1: Estimating the gradient of the $k$-subset sampling method*
>
> To clarify, the gradient required for GS, STGS, and SIMPLE is the gradient of the downstream function $f$. We do not differentiate the methods themselves. The key difference between these methods and our proposed method is that it does not require the downstream function’s gradient and can thus be used in a broader setting.
>
> *Interpretation 2: Using estimated gradients of $f$*
>
> This is interesting. In principle, it is possible to combine the methods requiring gradients with a gradient estimator for the pathwise gradient if it is unavailable. However, we are not aware of any such attempts, and trivial solutions would be costly and likely prone to error propagation due to using an estimate to compute another.
>
> **Q3. Interpreting Figure 5**
>
> Figure 5 highlights the importance and effectiveness of variance reduction. In previous work score function estimators have been dismissed due to their high variance, and Figure 5 shows us why — variance reduction is crucial for constructing a score function estimator. Our proposed estimator SFESS+VR shows that variance reduction (along with our proposed method for efficiently computing the score function) makes score function estimators a viable option for gradient estimation in $k$-subset sampling. The viability of score function estimators with variance reduction is the main message of the paper.
>
> **References**
>
> [1] Van Gorp, Hans, Iris Huijben, Bastiaan S. Veeling, Nicola Pezzotti, and Ruud JG Van Sloun. Active deep probabilistic subsampling. In International Conference on Machine Learning, 2021.
>
> [2] Zena M. Hira 1, Duncan F. Gillies. A Review of Feature Selection and Feature Extraction Methods Applied on Microarray Data. In Adv Bioinformatics, 2015.
>
> [3] Jianbo Chen, Le Song, Martin Wainwright, and Michael Jordan. Learning to Explain: An Information-Theoretic Perspective on Model Interpretation. In International Conference on Machine Learning, 2018.

---

### Official Review · Reviewer_W34X · 2024-10-31

**Soundness:** 3
**Presentation:** 3
**Contribution:** 2
**Rating:** 5
**Confidence:** 4

**Summary:**

This paper introduces a method for computing the score function of the k-subset distribution using a discrete Fourier transform and employs control variates to address the high variance of the vanilla score function estimator.

**Strengths:**

SFESS broadens the potential applications of k-subset sampling, particularly in scenarios where the downstream model’s gradient is unavailable. The algorithm’s effectiveness is demonstrated through several experiments.

**Weaknesses:**

This paper appears to fall slightly below the standard expected at ICLR for several reasons:

1. A primary concern is the lack of a solid theoretical foundation; the paper focuses mainly on numerical results without a deeper mathematical analysis. For example, there is no mathematical description or convergence rate guarantee for Algorithm 2.

2. In terms of experiments, the explanations of the architecture design and choice of hyperparameters require more clarity and justification. I will outline these concerns in greater detail in the questions below.

3. Additionally, the paper would benefit from further polishing. For instance, on line 204, p_{theta}(b) is introduced without prior definition. On line 206, the phrase should be “Naively.” Also, on line 221, the operation ArgTopK(log(theta+g), k) would benefit from further explanation.

**Questions:**

1. Choice of Hyperparameter k: How do you select the hyperparameter k for different tasks? How does performance vary if k is adjusted? A deeper discussion on the choice of k would be beneficial.

2. Selection of Random Seeds: How did you choose your random seeds? It would be helpful to list the specific seeds used to facilitate reproducibility and to confirm that your choices were not made deliberately.

3. Performance Comparisons in Tasks: In the VAE task, it appears that SIMPLE outperforms SFESS+VR (Table 3), while for the kNN task, GS and STGS perform better (Table 4). What is the intuition behind these results, and what advantages does SFESS+VR offer in these contexts?

I would be willing to increase the score if the questions above are thoroughly addressed and if more in-depth theoretical results are provided.

---

> ### Author Response · Authors · 2024-11-21
> **Author Response to Reviewer W34X**
>
> Thank you for your valuable feedback. We address your comments and questions below.
>
> **W1: Theoretical foundation**
>
> Algorithm 2 is a standard application of stochastic gradient descent using the proposed gradient estimates. The algorithmic motif follows that of training a variational autoencoder [1] where gradient estimates are used within SGD (this is sometimes called doubly stochastic optimization due to the two sources of variance: the subsampling of data and the gradient estimates [2]). This optimization approach has become widespread in deep learning in the last decade. For example, the gradient estimators GS, STGS, IMLE, and SIMPLE were all proposed in this setting without proof of convergence. In the same vein, SFESS is a gradient estimator that we evaluate as part of this common approach.
>
> SFESS also inherits the theoretical foundation of score function estimators. Perhaps the most important property is its unbiased gradient, a proof of which is presented in Appendix A. Unbiased gradient estimates are a common assumption in the analysis of e.g. gradient descent algorithms which does not hold for GS, STGS, IMLE, or SIMPLE. As an example, Lemma 4.2 in Bottou et al. 2018 [3] shows the expected improvement between two iterates for continuously differentiable downstream functions $f$ with Lipschitz-continuous gradient $\nabla f$. For unbiased gradient estimators (like SFESS), the expected improvement is bounded depending on the learning rate, the Lipschitz constant of $\nabla f$, and the second moment of the gradient estimator — which motivates variance reduction to improve convergence.
>
> Finally, to clarify the wording in Algorithm 2: training until convergence refers to its usual meaning in deep learning and not a formal notion of convergence, i.e. training until some metric no longer improves significantly.
>
> **Q1: How was $k$ chosen?**
>
> Note that, the subset size $k$ is not chosen. It is given as a part of the problem definition (gradient estimation of $k$-subset sampling). As such, it exists for all other methods as well. Importantly, SFESS works for any $k$, unlike GS and STGS which deteriorate with larger $k$ (see line 427). As for the experimental setup, we compare all methods in various problem settings ($n$ and $k$) as shown in Tables 2-4.
>
> **Q2: How were the random seeds chosen?**
>
> The random seeds were not chosen deliberately. The random seeds $\\{0, 1, 2, 3, 4\\}$ were used for repeated experiments. We have made the code available through an anonymous repository in a separate comment. The code will also be made publicly available upon publication to facilitate reproducibility.
>
> **Q3: Performance comparisons**
>
> We would like to emphasize that we do not claim that SFESS provides superior performance to other methods in all settings. The paper proposes that score function estimators should not be dismissed as they produce comparable performance to state-of-the-art methods (GS, STGS, SIMPLE) in the showcased experiments while providing qualitative advantages as mentioned in W1 of response to Reviewer kBPo.
>
> To give some examples of the broader applicability of score function estimators, consider the following downstream functions that are not differentiable:
> - A black-box function or e.g. a network with quantized weights.
> - Directly optimizing a non-differentiable metric (e.g. accuracy).
> - A reinforcement learning environment with a non-differentiable reward function.
>
> **References**
>
> [1] Diederik P. Kingma and Max Welling. Auto-Encoding Variational Bayes. In International Conference on Learning Representations, 2014.
>
> [2] Titsias, Michalis, and Miguel Lázaro-Gredilla. Doubly stochastic variational Bayes for non-conjugate inference. In International Conference on Machine Learning, 2014.
>
> [3] Bottou, Léon, Frank E. Curtis, and Jorge Nocedal. Optimization Methods for Large-Scale Machine Learning. In SIAM Review, 60.2 223-311, 2018.

---

> > ### Comment · Reviewer_W34X · 2024-11-26
> > **Increase the score**
> >
> > Thanks to the author for their response and explanation. Based on the changes, I decided to increase my score to 5.

---

> > > ### Author Response · Authors · 2024-11-27
> > > **Remaining concerns**
> > >
> > > Thank you for your response and the revised score. We would be grateful if you could clarify what remaining concerns stop you from recommending accepting the paper. As the discussion period has been extended, we are happy to address any remaining questions and concerns.

---

### Official Review · Reviewer_kBPo · 2024-11-03

**Soundness:** 3
**Presentation:** 3
**Contribution:** 2
**Rating:** 5
**Confidence:** 3

**Summary:**

The paper proposes a black-box gradient estimator for $k$-subset sampling using a score function estimator.
In this approach, the discrete Fourier transform (DFT) is employed to compute the density function of the Poisson binomial distribution, and control variates is used to reduce variance.
The method is evaluated through multiple experiments, where it is compared against existing approaches.

**Strengths:**

1. The use of a score function to address the gradient estimation problem is straightforward yet effective. To compute the probability density of the Poisson binomial distribution, the paper leverages a closed-form expression based on the discrete Fourier transform (DFT). A variance reduction technique is then applied to enhance estimation efficiency.
2. The paper is well-written, providing a clear overview of the background, challenges, and gaps in the field, along with a detailed summary of $k$-subset sampling methods.
3. The proposed method is validated on both synthetic and real-world dataset.

**Weaknesses:**

1. While the paper proposes a score function estimator for $k$-subset sampling, Equation (4) appears to be a well-established technique, commonly known as the score function estimator (SFE) [1].
This suggests that the primary contribution may lie in the application of variance reduction within the Monte Carlo approximation.
My main concern is that the novelty of this work may be insufficient for acceptance at a top-tier conference main track.

2. SFESS does not appear to demonstrate a consistently superior performance in the experimental results compared to other methods (e.g., SIMPLE). Could you clarify the specific advantages of SFESS and provide more insight into scenarios where it outperforms alternative approaches?

[1]. Kareem Ahmed, Zhe Zeng, Mathias Niepert, and Guy Van den Broeck. SIMPLE: A Gradient Estimator for k-Subset Sampling. In International Conference on Learning Representations, 2023.

**Questions:**

Please see Weaknesses part.

---

> ### Author Response · Authors · 2024-11-21
> **Author Response to Reviewer kBPo**
>
> Thank you for your valuable feedback. We address your comments and questions below.
>
> **W1: Novelty of score function estimators**
>
> We agree with the reviewer that score function estimators are a well-established technique in general, which is why we would like to bridge this research gap for the specific task of $k$-subset sampling. Interestingly, prior works on $k$-subset sampling [1, 2, 3] mention score function estimators and their high variance or the cost of computing the score function. The following excerpts highlight this research gap:
>
> > Score-based methods such as REINFORCE [Williams, 1992] for estimating the gradient of such objectives typically have high variance. [1] (page 1)
>
> > SFE uses faithful samples and exact marginals (which is feasible only when m is very small) and converges much more slowly than the other methods [...] [2] (page 8)
>
> > [The score function estimator] is typically avoided due to its notoriously high variance, despite its apparent simplicity. [3] (page 3)
>
> Besides identifying this research gap, we address the two limitations above by (1) proposing an efficient method for computing the score function, and (2) using control variates to reduce the estimator's variance. We show that this novel combination of techniques results in comparable results to existing state-of-the-art methods on various tasks in sections 4.2 to 4.4 while providing valuable qualitative advantages: unbiased gradient estimates, applicability to non-differentiable $f$, and not requiring hyperparameter tuning (see Table 1).
>
> SFE in SIMPLE is an example of how score function estimators for $k$-subset sampling were not viable prior to our contributions. In SIMPLE. the score function was computed by enumerating all subsets in a toy problem (the problem adopted in section 4.1). This is not feasible for larger experiments, which is most likely why SFE was not included in SIMPLE's larger experiments (Tables 2 and 3 in [3]). In addition, its performance without variance reduction would have been poor (see SFESS vs. SFESS+VR in our experiments).
>
> **W2: Results compared to other methods**
>
> We would like to emphasize that we do *not* claim that SFESS provides superior performance to other methods in all settings. The paper proposes that score function estimators should not be dismissed as they produce comparable performance to state-of-the-art methods (GS, STGS, SIMPLE) in the showcased experiments while providing qualitative advantages as mentioned in response to W1 above.
>
> To give some examples of the broader applicability of score function estimators, consider the following downstream functions that are not differentiable:
> - A black-box function or e.g. a network with quantized weights.
> - Directly optimizing a non-differentiable metric (e.g. accuracy).
> - A reinforcement learning environment with a non-differentiable reward function.
>
> **References**
>
> [1] Sang Michael Xie and Stefano Ermon. Reparameterizable Subset Sampling via Continuous Relaxations. In International Joint Conference on Artificial Intelligence, 2019.
>
> [2] Mathias Niepert, Pasquale Minervini, and Luca Franceschi. Implicit MLE: Backpropagating
> Through Discrete Exponential Family Distributions. In Advances in Neural Information Pro-
> cessing Systems, 2021.
>
> [3] Kareem Ahmed, Zhe Zeng, Mathias Niepert, and Guy Van den Broeck. SIMPLE: A Gradient Estimator for k-Subset Sampling. In International Conference on Learning Representations, 2023.

---

### Author Response · Authors · 2024-11-29
**Reminder and a comment on experiments**

We thank all reviewers for their thoughts and feedback and would like to remind them that the discussion period is ending in just a few days. If our responses addressed your concerns and answered your questions, please consider revising your scores. If there are any remaining or follow-up questions, post them as soon as possible and we will do our best to answer them.

Regarding the performance comparison in experiments, we emphasize that the paper does not claim consistently superior performance to existing methods. Instead, we aim to show that our proposed method (using efficient score computation and variance reduction) makes the score function estimator a viable alternative to existing approximate pathwise and relaxed sampling methods. Beyond the quantitative comparison, consider that:
- Our proposed method provides qualitative advantages (see W1 in our response to Reviewer kBPo).
- Our proposed method is complementary to existing works. In fact, combining pathwise estimators and score function estimators (as was done for categorical samples by Tucker et al. 2017 [1]) is a possible direction for future work.

**References**

[1] George Tucker, Andriy Mnih, Chris J. Maddison, Dieterich Lawson, and Jascha Sohl-Dickstein. REBAR: Low-variance, unbiased gradient estimates for discrete latent variable models. In Neural Information Processing Systems, 2017.

---

### Meta-Review · Area_Chair_kiPh · 2024-12-20

**Metareview:**

This paper proposes a way to estimate the score function for k-subset sampling by using a discrete Fourier transform.

In my opinion, the core idea of this work comes from noting that (6) can be used for score estimation in the context of score estimation of k-subset sampling. The authors showcased the advantage of such a method (which IMO are a consequence of its simplicity): exact samples, unbiased gradient, works for $f$ non-differentiable and not have any hyperparameter to be tuned). For the *tuning-free* aspect, I would like to mention that it is not completely accurate since the size of the mini-batch of the control variate is a hyperparameter whose impact is investigated in Figure 3.

Overall, I believe this paper proposes a valuable contribution to our field and recommend this paper to be accepted (though I would be okay if the Senior AC do not follow my recommendation as this paper is probably the most borderline paper I recommended for acceptance)

**Additional Comments On Reviewer Discussion:**

Reviewer W34X changed their rating from 3 to 5 after the discussion period.

The other reviewers did not engage in the discussion or even acknowledge the rebuttal even after I sent them a specific message, so I decided not to take too much their review in to account.

After reading the reviews and the paper I believe this paper could be accepted (though it is very borderline)

---

### Decision · Program_Chairs · 2025-01-22

Accept (Poster)